# Do Global Value Chains Make Firms More Vulnerable to Trade Shocks?—Evidence from Manufacturing Firms in Sweden

**A. M. M. Shahiduzzaman Quoreshi \*** and **Trudy-Ann Stone \***

Department of Industrial Economics, Blekinge Institute of Technology, SE-371 79 Karlskrona, Sweden
\* Correspondence: shahiduzzaman.quoreshi@bth.se (A.M.M.S.Q.); trudy-ann.stone@bth.se (T.-A.S.)

**Abstract:** This paper examines the effect of the Global Financial Crisis on manufacturing firms in Sweden by analyzing the effect of trade exposure on firm performance. This study examines the decline in international trade during the global financial crisis by focusing on the relationship between global production linkages and firm performance. The trade exposure at the firm and industry levels were measured to assess the direct and indirect effects of the crisis on firm performance. Robust evidence was found of a negative relationship between trade exposure and the firms' sales and value-added growth during the crisis. In addition, it was found that higher export dependence was associated with lower sales growth during the crisis. Our results also show that the effect of the decline in the external demand on firm performance depends on the international input-output linkages. In particular, industries that are upstream in the value chain experienced a less severe decline in performance during the crisis.

**Keywords:** financial crisis; firm performance; exports; global value chains

**JEL Classification:** F14; F44; F6; L25

## 1. Introduction

The Global Financial Crisis (GFC) of 2008–2009 has resulted in one of the most severe economic downturns in recent times and triggered economic recessions in many countries across the world. Numerous studies have been conducted on the impact of GFC on the macroeconomy and microeconomy. The studies cover a wide range of areas, e.g., international trade, foreign direct investment, the connectedness between commodity markets, the currency markets and stock markets, innovation as well as volatilities (Jimborean and Kelber 2017; Batuo et al. 2017; Zhang and Broadstock 2018; Quoreshi et al. 2019). The literature on the macroeconomy focuses on the macroeconomic effects of GFC, such as understanding the causes of business cycle synchronization (Di Giovanni and Levchenko 2010; Baldwin and Evenett 2009; Bussière et al. 2013). However, the microeconomic effects of the global recession are less well understood. The macroeconomic models predict that, under certain conditions, exchange rate depreciations following the financial crises may lead to either an expansion or reduction in exports (Kim et al. 2015). However, these models do not account for the role of global production linkages between firms. Another branch of literature identifies financial market contagion as the primary channel through which the crisis stimulated a downturn in trade flows. Mollah et al. (2016) investigated whether contagion spread via the equity or credit markets by studying the correlation in financial returns for each country with U.S. financial returns. The authors show that bank risk transfer was responsible for the spread of the crisis from the U.S. to other countries. The crisis also meant that industries and the firms that had been more reliant on external credit were more adversely affected

(Foley and Manova 2015). In a seminal paper, Chor and Manova (2012) studied the impact of the crisis on the export performance at the industry level. They found that industries that relied on external credit fared worse during the crisis period. Chor and Manova (2012) also showed that this decline was worse in industries exporting to countries with less developed financial markets. The crisis resulted in tightening credit conditions in exporting countries, which made it more difficult for firms to secure working capital needed to produce as well as export (Chor and Manova 2012).

During this period, global trade experienced the steepest decline since the Great Depression (Bussière et al. 2013; Brakman et al. 2015). One distinguishing feature of the 2008–2009 crisis is that the contraction in trade has been more severe than the initial decline in output (Bussière et al. 2013). The decline in world exports is four times larger than the initial decrease in aggregate demand, as imports and exports deteriorate simultaneously across all countries (Baldwin and Evenett 2009; Bems et al. 2013). These developments have led to renewed interest in understanding the relationship[1] between trade flows and business cycle synchronization between countries, with particular emphasis being placed on the role of global production linkages in explaining why the crisis resulted in an even greater decline in trade (Levchenko et al. 2010; Behrens et al. 2013; Baldwin and Evenett 2009). One key finding is that production linkages may transmit a business cycle downturn across countries as a decline in demand for final goods in one country suppresses demand for foreign intermediate goods along the value chain (Baldwin and Evenett 2009; Freund 2009).

This paper examines one aspect of the decline in international trade during the global financial crisis by focusing on the relationship between global production linkages and firm performance. By being more integrated in global value chains, firms are more exposed to fluctuations in foreign demand and are expected to be more sensitive to external shocks. The effect of the recent economic crisis is tested on three main measures on firm performance: Export growth, domestic sales growth and value-added growth. The precise effect of the crisis on a firms' imports and exports may vary according to the extent to which a particular industry is dependent on a globally dispersed supply chain (Di Giovanni and Levchenko 2010). For instance, the automotive and the electronics manufacturing industries account for the largest share of trade in intermediate input trade. These two industries experienced the steepest decline in trade flows during the crisis years (Baldwin and Evenett 2009; Kawakami and Sturgeon 2010).

By using export growth and domestic sales growth as our main measures of firm performance, this paper can shed light on the direct and indirect effects of the global financial crisis on firms. It is argued that the crisis directly influences firm performance through its impact on export sales. Export sales may decline on two margins: The volume of export sales within each market (intensive margin[2]) and the number of markets a firm serves (extensive margin[3]). The crisis may also have a direct effect on firm performance by influencing the price of imports. A decline in the price of imported inputs may affect firm outcomes since a reduction in import prices can lower the cost of intermediate goods and boost performance for firms relying on imported inputs. Alternatively, lower import prices[4] for final goods may raise competition in the domestic market and reduce the market share of import-competing

---

[1] Early research on the relative contribution of finance and trade to business cycle synchronization includes Van Rijckeghem and Weder (2001), Glick and Rose (1999) and Frankel and Rose (1998).

[2] Behrens et al. (2013) show that Belgian exports declined more noticeably on the intensive margin, and this decline was more visible for firms that are more heavily indebted. Similarly, Paravisini et al. (2014) find that, in Peru, the intensive margin of exports declined more severely for firms more sensitive to credit shocks.

[3] Research on the effect of the crisis on the extensive margin of exports shows that export sales decline as firms exited certain markets during the 2008–2009 period. Martin et al. (2013) show that firms were less resilient during the crisis as both export survival and export growth rates declined during 2008–2009.

[4] Looking at Argentine exports, Chen and Juvenal (2015) investigate the effect of the crisis on the quality composition of Argentine exports. They find that the composition of exports shifted towards lower quality products during the crisis. In a similar vein, Bems and di Giovanni (2016) use scanner-level data from Latvia to document changes in consumption and find that consumers substituted imported goods for domestic goods and that this change accounts for 26 percent of the observed decline in imports.

domestic firms (Forbes 2002). The indirect effect of the crisis is manifested in the domestic market and operates through the input-output linkages between domestic firms and exporters. (Cravino and Levchenko 2016; Levchenko et al. 2010. The indirect effect may also occur as a result of the complementary effect of a firm's foreign operations on domestic sales. Brakman et al. (2015) found a positive effect of exports on domestic sales. This paper focuses on the importance of the input-output linkages and how they influenced firm performance during the 2008–2009 crisis.

The effect of global production linkages was analyzed by regressing firm exports and domestic sales growth on pre-crisis measures of trade exposure. Three main variables measuring the extent to which firms are exposed to trade were constructed. These include the firm-level trade intensity, industry-level exposure to trade and a measure of Swedish industries' position in the global value chain. In addition, a host of firm and industry characteristics have been controlled for, such as the skill level of workers, firm size, and the degree of industrial concentration using the Herfindahl Hirschman index as well as industry dummies. The data on Swedish firms' monthly exports, sales and value added for the period 2007–2014 were used. The dataset covers all firms in the Swedish manufacturing sector that declare monthly value added tax (VAT) and payroll taxes. Firm-level data were combined with industry measures of global value chain participation to investigate the role of input-output linkages between Swedish and foreign industries and how these linkages influenced firm performance during the crisis period.

The estimation results support the argument that global production linkages are a key mechanism through which the global financial crisis affected the firms' export, sales and value-added growth. First, it was found that the crisis had a negative effect on the firms' export and sales growth. Moreover, the decline in the firms' export growth was more severe than the decline in domestic sales growth. This indicates that greater exposure to trade through deeper participation in global value chains may make firms more sensitive to changes in external demand. Second, it was found that firms that operated in upstream industries experienced a lower decline in export growth, domestic sales growth and value-added growth.

While vertical linkages explain much of the decline in trade, the extent to which these linkages account for the observed reduction in trade in the wake of the global recession is widely debated (Duval et al. 2016; Ng 2010). There are two competing hypotheses explaining this falloff. One branch of literature argues that global trade collapsed as the crisis led to a decline in the demand for imports during the 2008–2009 period (Bems et al. 2010). Alessandria et al. (2011) examined the effect of the changes in demand on trade flows and put forward that inventory adjustments during the crisis led to a decrease in the demand for imports. The crisis resulted in a decline in consumer demand which encouraged firms to cut inventory on hand and fill orders from their existing stock of inventories in response (Alessandria et al. 2011). This then led to a decline in the demand for imports during the global economic recession.

This paper finds that evidence supporting the claim that global production linkages is one key mechanism through which the sudden decrease in external demand resulted in a decline in the firms' sales and export performance. This paper contributes to the current debate by highlighting the effect of production linkages in determining the effect of external shocks on firm performance. Most micro-level studies examine the relationship between a firms' reliance on external finance and firm performance during the crisis (Amiti and Weinstein 2011). These studies also typically test the effect of exchange rate depreciations on firm performance (Ekholm et al. 2012; Forbes 2002). This paper differs from existing studies in that, while the importance of financial vulnerability during economic recessions is acknowledged, it is argued that the increasing interconnectedness in global production adds a new dimension to the firms' responsiveness to financial crises. This paper is most closely related to Claessens et al. (2012) who found that trade linkages explain much of the decline in firm performance during the global financial crisis and that financial vulnerability is a less compelling explanation. Claessens et al. (2012) used data on publicly listed manufacturing firms in their analysis.

This paper examines the case for all manufacturing firms in Sweden, which allows the analysis of the heterogeneous effects of the decline in trade on firms.

In Section 2, an empirical analysis is presented and these results are discussed in Section 3. Finally, in Section 4 the main conclusions are presented.

## 2. Data and Empirical Approach

### 2.1. Empirical Approach

In this section, the main empirical approach is presented. Firm performance is measured denoting $y_{ijt}$ for sales growth or value-added growth where $i$ and $j$ represent firm respective industry at time period $t$. Firstly, the baseline specification is expressed which estimates the relationship between the effect of international linkages on firm performance and how this is affected by the crisis as:

$$\Delta y_{ijt} = \alpha + \beta_1 \Delta trade_{ij1} + \beta_2 crisis_t + \beta_3 \Delta trade_{ij1} \times crisis_t + \delta \Delta Z_{ijt} + \mu_t + \varepsilon_{ijt} \qquad (1)$$

where the dependent variable, $\Delta y_{ijt}$, and the independent variable $trade_{ij1}$ denotes firm-level pre-crisis exposure to international trade during 2007 and is defined as:

$$trade_{j1} = \left(import_{ij1}/export_{ij1}\right)/sales_{ij1}$$

Equation (1) also controls for firm characteristics, denoted by the matrix $Z_{ijt}$, such as firm size, a dummy variable indicating whether the firm is a multinational as well as a dummy variable indicating whether the firm has ever exported during the period 2007–2014. The monthly wage growth per worker is also included as a measure of the skill level of the firm's workforce which controls for certain firm-specific characteristics, such as productivity that may be correlated with firm performance and may therefore disguise the true effect of the crisis. Finally, the variable μt is included, which represents year dummies to control for factors such as changes in input prices that are common to all firms during a particular year.

Equation (1) answers the following question: What is the average effect of the crisis across the firms in an industry? The main purpose is to analyze whether value chain integration, measured at the industry level, influences the effect of the financial crisis on sales and value-added growth. By using the data on the firm's international activities may underestimate the true effect of trade exposure since the activities of other firms within each industry should also be taken into account. Two main predictions are tested: During the global financial crisis, the firms in sectors with stronger production linkages experienced larger declines in total sales growth. The second prediction states that during the global financial crisis, the firms in sectors with stronger global production linkages experienced larger declines in value-added growth.

To this end, the interaction between trade exposure at the industry level and the financial crisis dummy is included. The new specification is set out in Equation (2).

$$\Delta y_{ijt} = \alpha + \beta \Delta trade_{j1} + \beta_2 crisis_t + \beta_3 \Delta trade_{j1} \times crisis_t + \delta \Delta Z_{ijt} + \mu_j + \mu_t + \varepsilon_{ijt} \qquad (2)$$

where the dependent variable $\Delta y_{ijt}$ denotes changes in firm performance as in Equation (2). The variable tradej1 is included, which represents pre-crisis trade exposure at the industry-level and is measured using the natural logarithm of industry-level imports or exports. Alternatively, a measure of industry-level exposure to trade is constructed and defined as:

$$trade_{exposure_{j1}} = total\ production_{j1} - Exports_{j1} + Imports_{j1}$$

This measure of trade exposure accounts for changes in each industry's business cycle by accounting for monthly total sales at the industry level. Using pre-crisis variables as an independent variable helps us overcome potential problems associated with endogeneity (Kim et al. 2015). Pre-crisis

characteristics were therefore controlled for using data for the corresponding 12-month period up to September 2008. The industry's Herfindahl index is also included to control for the degree of concentration in each industry. The variable $Z_{ijt}$ denotes firm characteristics such as wage per employee, firm size, and dummies indicating MNE and exporter status. The lag of the dependent variable as a regressor is also included to account for serial correlation. The μj represents one-digit industry fixed effects to control for the effect of unobserved industry characteristics. Finally, $\mu_t$ represents year fixed effects. As in Equation (1), including the sector and year fixed effects controls for factors such as changes in input prices.

Equation (2) analyzes the effect of the crisis on the relationship between firm performance and global production linkages. This specification assists in the understanding of the relationship between trade exposure and sales (value-added) growth at the firm level, and how this relationship was influenced by the crisis. The coefficient of interest is $\beta_3$ which measures how trade exposure influences the impact of the crisis on firm performance. A negative and statistically significant point estimate for $\beta_3$ suggests that there is a negative relationship between trade exposure and firm performance during the crisis.

## 2.2. Data Description

To estimate the effects of the crisis on firm performance, a unique dataset on monthly tax returns for Swedish firms was used. The dataset covers all firms in Sweden (over 1,500,000 firms).It was obtained from Statistics Sweden (SCB) and the Swedish tax agency for the period 2007–2014. The dataset includes monthly values for sales, value-added, imports and exports. Other firm-level data available in the dataset include the data on firm characteristics such as size, ownership type as well as the municipality in which the firm is located. The data on monthly financial returns were used for the following reasons. First, high-frequency data provides a richer, more detailed level of analysis on the responses of firms to external shocks. More aggregate data at the annual level may wipe out periodic variation in firm performance. Second, monthly financial returns data also allows the analysis of how variations in macroeconomic variables such as exchange rates help to shape the firm-level outcomes. These effects are more difficult to detect using aggregate data. Finally, our dataset also includes variables such as interest payments which allow the analysis of the effect of the crisis on the firms' cost of capital.

This paper defined the crisis period from September 2008 to December 2009 in accordance with (Mollah et al. 2016). All financial variables were deflated using the appropriate industry price deflator. The domestic price index was used for each Swedish Industry Code (SNI) three-digit industry to deflate the variables, domestic sales and wages. These data on industry-level domestic, import and export price indices also came from SCB. The corresponding export and import indices were used for each SNI three-digit industry to deflate data on the firms' exports and imports, respectively. The analysis was restricted to manufacturing firms and firms with zero sales for the entire period were removed. The final sample consisted of an unbalanced panel of 56,422 firms. Finally, the analysis used the data on the value chain position Swedish industries. These measures were calculated based on data from the World Input-Output Database and are obtained from Stone (2016).

Table 1 summarizes the basic descriptive statistics for firms in the sample. Approximately 60% of all manufacturing firms exported at least once between 2007 and 2014. Multinationals make up 7% of all firms in the sample. The firms were aggregated into 13 industries.

**Table 1.** Descriptive Statistics.

| Variable | Number of Observations | Mean | Std. Dev. | Min | Max |
|---|---|---|---|---|---|
| Log exports | 1,474,129 | 3.887 | 6.127 | 0 | 23.177 |
| Log imports | 418,566 | 12.159 | 2.509 | −0.193 | 22.216 |
| Log value added | 1,110,482 | 11.591 | 1.857 | 0.128 | 26.00 |
| Log sales | 1,474,129 | 10.083 | 6.065 | 0 | 23.404 |
| Log wage/worker | 1,474,131 | 9.190 | 2.554 | 0 | 15.06 |
| No. workers | 1,474,132 | 31.170 | 277.23 | 1 | 19,072 |
| Industry | 1,474,133 | 7.969 | 3.88 | 1 | 13 |
| Foreign MNE | 1,474,134 | 0.065 | 0.247 | 0 | 1 |
| Exporter | 1,474,135 | 0.60 | 0.48 | 0 | 1 |

Figure 1 shows the trends in median sales and value-added growth for all manufacturing industries. Both foreign and domestic sales growth declined during the period July 2008–January 2010, with the steepest decline occurring between June and August 2009. These trends are comparable to GDP growth for Sweden, which declined between the first quarter of 2008 and the third quarter of 2009.

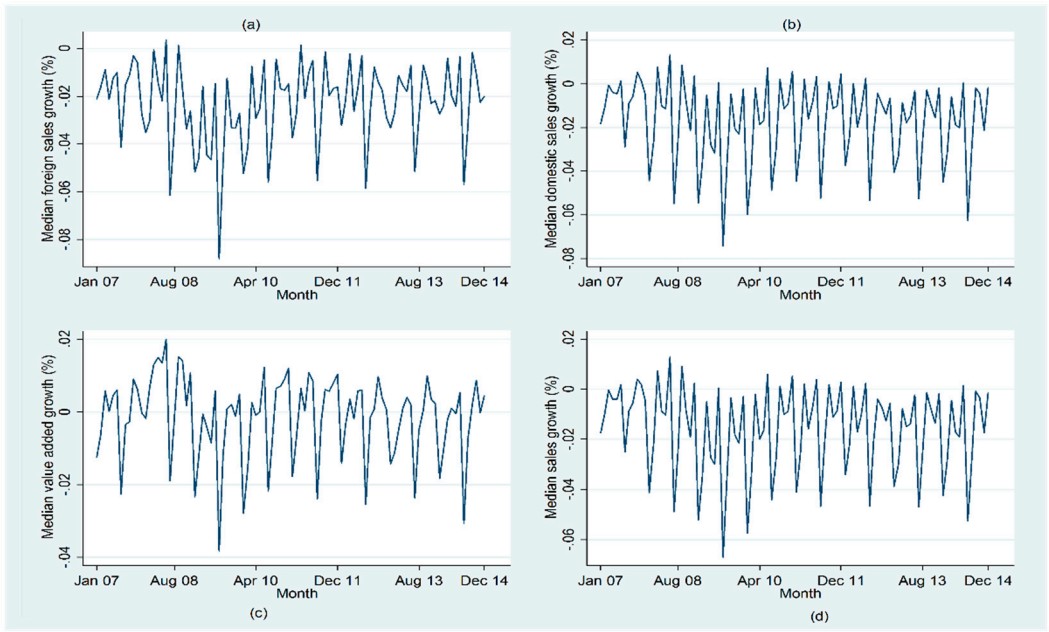

**Figure 1.** Median monthly sales and value-added growth for manufacturing firms (2007–2014). Note: (**a**,**b**) show the median foreign sales growth and median domestic sales growth. (**c**) shows median value-added growth while (**d**) shows median total sales growth.

Figure 2 shows quarterly GDP growth between 2006 and 2014. The grey bar indicates the recession period. During the fourth quarter of 2008, quarterly GDP fell by approximately 3.8 percent. These patterns in foreign sales growth also are comparable to trends in world trade flows. In Appendix A (Figure A1), 2009 features a steep decline in trade flows across the world and more so for Sweden. In 2009, world trade declined by 10% while Swedish trade flows decreased by 15% on an annual basis. As discussed in the introduction, the decline in GDP brought about an even steeper decline in world trade.

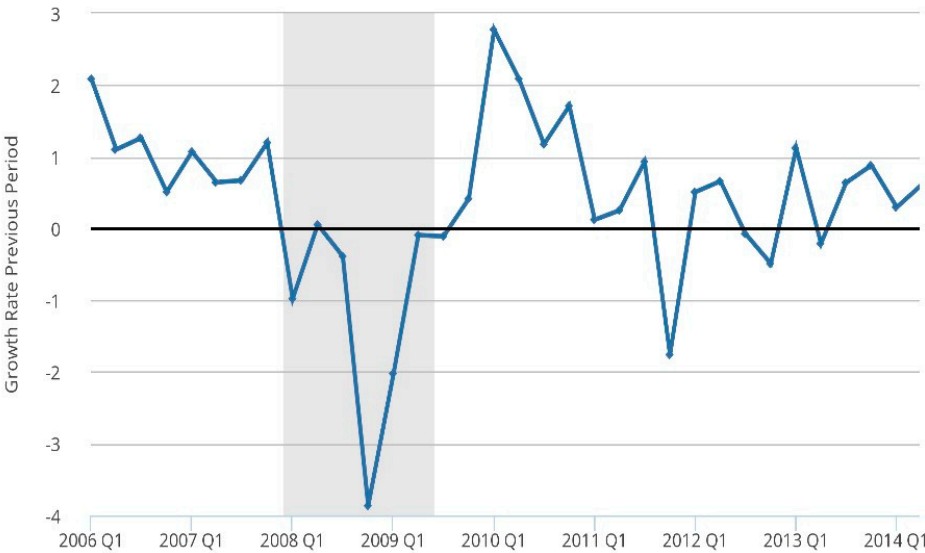

**Figure 2.** Quarterly GDP Growth for Sweden (2007–2014). Source: Federal Reserve Bank of St. Louis (FRED).

## 3. Results

In this section, the results of the regression analysis are discussed. Firstly, the results of the baseline specification are discussed. To analyze the effect of the crisis on firm performance, Equation (1) was estimated by regressing changes in sales and changes in value-added on a crisis dummy and lagged firm characteristics. Table 2 compares the results for the two measures of firm performance—sales growth and value-added growth. Columns 3 and 4 show that there is a negative relationship between our crisis variable and monthly sales and value-added growth, respectively. The results in Table 2 indicate an average decline in monthly sales and value-added growth by between 2% and 4% during the crisis. The variable, total sales growth was decomposed into foreign and domestic sales growth and it was found that there was a larger decline in domestic sales growth (3.4%) than foreign sales growth (1.6%).

Turning to the full results for Equation (1), the interaction variable for three measures of trade exposure were included. The results are presented in Table 3. Columns 1–3 show the effect of trade exposure on a firm's monthly foreign sales growth. A negative coefficient was obtained for the interaction term between monthly import growth at the firm level and the crisis dummy. The results suggest that there is a positive relationship between higher monthly import growth in the pre-crisis period and foreign sales growth overall. However, higher import growth is not a strong predictor of firm performance during the crisis. Column 2 shows the results when export growth is used as the measure of trade exposure. The results show that higher pre-crisis export exposure is positively related to foreign sales growth. However, firms with stronger export linkages in the pre-crisis period also experienced lower foreign sales during the crisis. This is seen in the negative and statistically significant coefficient for the interaction between the interaction between the crisis dummy and the firms' monthly pre-crisis export growth and the firm's overall foreign sales growth.

Column 3 reports the results when the measure of firm trade exposure presented in Section 3.1 is used. The interaction between the crisis dummy and firm exposure to trade is negative and significant. This result is interpreted as follows. The firms with a higher import and export intensity (import and exports as a share of total sales) in 2007 are more likely to experience a decline in export growth during the crisis. Our results also suggest that export growth is one of the main factors driving the negative impact of trade exposure on foreign sales growth.

**Table 2.** Monthly sales and value-added growth for manufacturing firms.

| Variables | Foreign Sales | Domestic Sales | Total Sales | Value Added |
|---|---|---|---|---|
| Crisis | −0.016 *** | −0.034 *** | −0.036 *** | −0.019 *** |
| | (0.004) | (0.006) | (0.006) | (0.002) |
| 1–9 employees | 0.081 *** | 0.419 *** | 0.427 *** | 0.028 *** |
| | (0.008) | (0.008) | (0.008) | (0.003) |
| 10–49 employees | 0.191 *** | 0.367 *** | 0.375 *** | 0.007 ** |
| | (0.008) | (0.009) | (0.009) | (0.003) |
| 50–249 employees | 0.024 *** | 0.024 ** | 0.026 *** | −0.001 |
| | (0.009) | (0.009) | (0.009) | (0.003) |
| Wage per worker | 0.013 *** | 0.057 *** | 0.057 *** | 0.025 *** |
| | (0.002) | (0.004) | (0.004) | (0.001) |
| HHI | −2.175 *** | −5.351 *** | −5.412 *** | −0.252 *** |
| | (0.115) | (0.257) | (0.260) | (0.021) |
| Constant | 0.077 *** | 0.191 *** | 0.187 *** | 0.019 *** |
| | (0.008) | (0.010) | (0.010) | (0.003) |
| Observations | 1,198,584 | 1,198,584 | 1,198,584 | 762,764 |
| R-squared | 0.221 | 0.214 | 0.213 | 0.220 |
| INDUSTRY FE | YES | YES | YES | YES |

Note: The sample includes all manufacturing firms with at least one employee. Lagged dependent variables are included in estimation but not reported in the table. Robust standard errors clustered at firm × year level where *** denotes $p < 0.01$ and ** $p < 0.05$.

Columns 4, 5 and 6 present the results for domestic sales growth. Here, it can be seen that foreign sales and domestic sales growth show similar responses to the crisis. Column 4 shows that import dependence at the firm level is negatively related to the firms' domestic sales growth. Similar results are obtained when export growth is used as the measure of trade exposure. In Column 6, a positive coefficient for our interaction variable has been obtained. This suggests that firms with higher import and export intensity are more likely to report positive domestic sales growth during the crisis. In Columns 10, 11 and 12, the results for value added growth are reported. A positive effect of trade exposure in 2007 and value-added growth during the crisis was found.

The results in Table 3 also reveal that firms that pay higher wages are more resilient. This is seen in the positive point estimate for the variable wage per employee. These results suggest that having a higher share of skilled employees is positively with associated higher sales and value-added growth. In addition, the results show that operating in more concentrated industries is negatively related to firm performance.

The effect for exporters and non-exporters were compared to determine whether international linkages influence firm performance during the crisis. These results are provided in the Appendix A (Tables A1 and A2). A wealth of research shows that exporters are larger and more productive than non-exporters (Bernard et al. 2007). This productivity advantage may make exporters more resilient to adverse macroeconomic conditions. A decline in demand in one market may influence firms to reorganize internally and increase sales in other markets, thereby making firms less responsive to recessions. The results showed that exporting manufacturing firms were negatively affected by the financial crisis. That is, the decline in sales growth was more severe for exporters. Similar results were obtained when value-added growth was used as the dependent variable. These results suggest that greater trade exposure negatively influenced firm performance during the crisis.

**Table 3.** The financial crisis and firm performance using the firm's own exposure to trade.

| Variables | (1) Foreign Sales | (2) Foreign Sales | (3) Foreign Sales | (4) Domestic Sales | (5) Domestic Sales | (6) Domestic Sales | (7) Total Sales | (8) Total Sales | (9) Total Sales | (10) Value Added | (11) Value Added | (12) Value Added |
|---|---|---|---|---|---|---|---|---|---|---|---|---|
| Crisis | 0.012 (0.022) | −0.040 (0.025) | −0.028 (0.025) | 0.020 (0.029) | 0.024 (0.025) | 0.025 (0.024) | 0.016 (0.028) | 0.015 (0.026) | 0.018 (0.025) | 0.012 (0.022) | −0.017 *** (0.006) | −0.016 *** (0.006) |
| HHI | −10.544 ** (4.667) | −12.731 ** (5.343) | −13.122 ** (5.153) | −15.848 ** (7.041) | −15.074 ** (6.475) | −15.373 ** (6.136) | −15.933 ** (7.035) | −15.250 ** (6.570) | −15.414 ** (6.193) | −10.544 ** (4.667) | −1.034 *** (0.224) | −1.083 *** (0.244) |
| Wage per worker | 0.064 *** (0.015) | 0.078 *** (0.016) | 0.081 *** (0.019) | 0.074 *** (0.015) | 0.074 *** (0.015) | 0.078 *** (0.018) | 0.084 *** (0.016) | 0.073 *** (0.016) | 0.073 *** (0.019) | 0.064 *** (0.015) | 0.022 *** (0.004) | 0.022 *** (0.005) |
| Import 2007 | 0.023 * (0.013) | | | 0.023 (0.023) | | | 0.011 (0.020) | | | 0.023 * (0.013) | | |
| Crisis × import 2007 | −0.018 (0.015) | | | −0.085 *** (0.019) | | | −0.065 *** (0.015) | | | −0.018 (0.015) | | |
| Exports 2007 | | 0.202 *** (0.029) | | | 0.022 (0.027) | | | 0.052 * (0.027) | | | 0.054 *** (0.006) | |
| Crisis × Exports 2007 | | −0.157 *** (0.028) | | | −0.037 * (0.021) | | | −0.080 *** (0.022) | | | 0.004 (0.007) | |
| Exposure 2007 | | | 0.079 ** (0.036) | | | −0.033 (0.038) | | | −0.006 (0.037) | | | −0.043 *** (0.005) |
| Crisis × Exposure 2007 | | | −0.046 ** (0.023) | | | 0.065 *** (0.019) | | | 0.033 * (0.018) | | | 0.027 *** (0.005) |
| Constant | 0.089 (0.492) | 0.382 (0.453) | −0.298 (0.996) | 1.019 *** (0.341) | 1.120 *** (0.347) | 1.209 *** (0.271) | 1.332 *** (0.300) | 1.121 *** (0.321) | 1.197 *** (0.247) | 0.089 (0.492) | 0.237 *** (0.046) | 0.462 *** (0.055) |
| Observations | 230,165 | 251,690 | 210,356 | 230,165 | 251,690 | 210,356 | 230,165 | 251,690 | 210,356 | 230,165 | 198,648 | 169,988 |
| R-squared | 0.245 | 0.228 | 0.232 | 0.143 | 0.241 | 0.246 | 0.252 | 0.240 | 0.244 | 0.245 | 0.242 | 0.243 |
| INDUSTRY FE | YES | YES | YES | YES | YES | YES | YES | YES | YES | YES | YES | YES |

Note: The sample includes all manufacturing firms with at least one employee. Lagged dependent variables and a categorical variable representing four firm size groups are included in estimation but not reported in the table. Robust standard errors clustered at 3-digit SNI industry level where *** denotes $p < 0.01$, ** $p < 0.05$, and * $p < 0.1$.

### 3.1. Industry Exposure to Trade and Firm Performance

The baseline specification confirms our hypothesis that the decline in firm performance during the crisis varies between firms according to the extent to which the firm participates in international trade. In this subsection, this heterogeneity in firm performance is explored and the effect of the crisis on the relationship between trade linkages and sales growth is examined. To investigate the effect of value chain integration on the responsiveness of firms to trade shocks, Equation (2) is estimated and the results are reported in Table 4. Industry level trade exposure defined in Section 3.1 was used since a firm's exposure to trade depends to a large extent on the industry in which it operates. Column 1 shows that there is a negative relationship between sales growth and the interaction between industry trade exposure and the crisis dummy. Columns 3 and 4 show the results for domestic and foreign sales growth, respectively.

**Table 4.** Firm performance and industry exposure to trade.

| Variables | (1) | (2) | (3) | (4) |
|---|---|---|---|---|
| | Δ Total Sales | Δ Value Added | Δ Domestic Sales | Δ Foreign Sales |
| Crisis | −0.147 *** | −0.039 *** | −0.133 *** | −0.072 *** |
| | (0.019) | (0.005) | (0.019) | (0.013) |
| Foreign MNE | −0.139 *** | −0.002 | −0.133 *** | −0.046 *** |
| | (0.014) | (0.002) | (0.014) | (0.008) |
| Δ Wage per worker | 0.051 *** | 0.025 *** | 0.051 *** | 0.011 *** |
| | (0.003) | (0.001) | (0.003) | (0.002) |
| Δ Industry exposure (t − 1) | 0.988 *** | −0.045 *** | 0.965 *** | 0.248 *** |
| | (0.012) | (0.004) | (0.012) | (0.009) |
| Crisis × Industry Exposure | −0.461 *** | 0.066 *** | −0.459 *** | −0.140 *** |
| | (0.026) | (0.011) | (0.026) | (0.022) |
| 1–9 employees | 0.290 *** | 0.028 *** | 0.284 *** | 0.050 *** |
| | (0.021) | (0.003) | (0.021) | (0.010) |
| 10–49 employees | 0.272 *** | 0.010 *** | 0.267 *** | 0.168 *** |
| | (0.022) | (0.003) | (0.022) | (0.011) |
| 50–249 employees | −0.032 | 0.001 | −0.032 | 0.002 |
| | (0.022) | (0.003) | (0.022) | (0.011) |
| HHI | −4.219 *** | −0.115 *** | −4.192 *** | −1.347 *** |
| | (0.027) | (0.002) | (0.027) | (0.014) |
| Constant | −16.142 *** | −0.420 *** | −16.033 *** | −5.207 *** |
| | (0.109) | (0.009) | (0.108) | (0.055) |
| Observations | 1,198,584 | 762,764 | 1,198,584 | 1,198,584 |
| R-squared | 0.298 | 0.223 | 0.298 | 0.242 |
| YEAR FE | YES | YES | YES | YES |
| INDUSTRY FE | YES | YES | YES | YES |
| FIRM FE | NO | NO | NO | NO |

Note: Lagged dependent variable included but not reported. Robust standard errors clustered at the firm × year level in parentheses where *** denotes $p < 0.01$.

It was found that greater trade exposure in 2007 was negatively related to export growth during the crisis. This is evidenced by the negative point estimate for the interaction between the crisis dummy and industry level exposure. The findings reveal that industry-level exposure has a greater effect on export growth than on domestic sales growth. While these results are intuitive, they also provide evidence that a firm's performance on the domestic market is influenced by its exposure to international trade. In Column 2, value-added growth is positively related to firm exposure to trade. These results may indicate that firms that are more exposed to trade may be able to reorganize internally during the crisis. Ekholm et al. (2012) found that sudden shocks, such as real exchange rate changes, are associated with higher productivity for firms that are relatively more exposed to trade.

### 3.2. Indirect Effects of the Crisis on Firm Performance

Thus far, our findings point to a negative relationship between trade exposure and sales growth during the global financial crisis. Our findings also suggest that the decline in the firms' export sales contributed to most of the observed reduction in firm performance. The role of interfirm linkages during the crisis is now considered and it is argued that sales and value-added growth also depends on the firm's backward and forward linkages[5]. The effect of the value chain position of Swedish industries on firms' sales and value-added growth is investigated. The value chain position may determine the firms' ability to adjust output, sell existing stock or to find new markets. The firms that specialize in upstream industries produce goods that are used as production inputs and may be constrained by the lack of alternative markets. For instance, if their output is tailored to the purchasing firm's requirements, a decline in demand may leave the firm with excess capacity and limited options to seek new market. On the other hand, since Swedish firms produce output that is closer to the final customer (i.e., downstream the value chain), a decline in demand may have a strong effect on industries that are downstream the value chain.

To test the effect of the value chain position, each industry's upstreamness in the global value chain in 2007 was controlled for using the measure proposed in Fally (2012). The firms were divided into two samples: The industries with upstreamness scores above the median and the industries with upstreamness scores below the median. A high upstreamness score implies that the firm produces outputs that are used as inputs in the production processes. The results are reported in Table 5. Columns 1–3 reports the results for sales and value-added growth for industries with a high upstreamness score (greater than the median). It was found that firms that operate in upstream industries reported a less severe decline in sales and value-added growth during the crisis. These findings suggest that the effect of trade shocks on firm performance depends not only on the extent to which a firm is exposed to trade, but also the on the firm's position in global value chains.

---

[5]　Levchenko et al. (2010) find that value chain linkages between countries contributed to a significant share of the decline in trade.

**Table 5.** The effect of the global financial crisis on firm performance: the role of value chain position.

| Variables | (1) Foreign Sales | (2) Domestic Sales | (3) Value Added | (4) Foreign Sales | (5) Domestic Sales | (6) Value Added |
|---|---|---|---|---|---|---|
| Crisis | −0.177 *** | −0.467 *** | −0.047 *** | −0.301 *** | −0.944 *** | −0.062 *** |
| | (0.016) | (0.015) | (0.007) | (0.019) | (0.018) | (0.008) |
| HHI | −0.318 *** | −0.806 *** | −0.022 *** | −0.313 *** | −1.215 *** | −0.028 *** |
| | (0.005) | (0.007) | (0.001) | (0.006) | (0.015) | (0.001) |
| Δ Wage per worker | 0.012 *** | 0.056 *** | 0.027 *** | 0.014 *** | 0.058 *** | 0.022 *** |
| | (0.002) | (0.005) | (0.001) | (0.002) | (0.006) | (0.002) |
| 1–9 employees | 0.114 *** | 0.496 *** | 0.038 *** | −0.236 *** | −0.869 *** | −0.010 |
| | (0.011) | (0.018) | (0.004) | (0.021) | (0.064) | (0.006) |
| 10–49 employees | 0.203 *** | 0.402 *** | 0.013 *** | −0.091 *** | −0.830 *** | −0.022 *** |
| | (0.012) | (0.018) | (0.004) | (0.021) | (0.064) | (0.006) |
| 50–249 employees | 0.021 | 0.021 | 0.005 | −0.118 *** | −0.586 *** | −0.023 *** |
| | (0.013) | (0.020) | (0.004) | (0.023) | (0.068) | (0.007) |
| Constant | −1.520 *** | −3.976 *** | −0.096 *** | −1.096 *** | −4.123 *** | −0.087 *** |
| | (0.025) | (0.039) | (0.006) | (0.029) | (0.078) | (0.008) |
| Observations | 729,634 | 729,634 | 473,354 | 468,950 | 468,950 | 289,410 |
| R-squared | 0.223 | 0.225 | 0.216 | 0.229 | 0.247 | 0.228 |
| YEAR FE | YES | YES | YES | YES | YES | YES |

Note: Lagged dependent variable included but not reported. Robust standard errors clustered at the firm × year level. Robust standard errors clustered at the firm × year level reported in parentheses where *** denotes $p < 0.01$.

## 4. Conclusions

Global production linkages have led to an increase in business cycle synchronization between countries and is one of the main factors explaining why the 2008–2009 financial crisis resulted in a collapse in world trade and severe economic recessions across many countries. However, most studies focus on the aggregate effect of the crisis on an international trade level. By studying how firms respond to external shocks, this paper is one of a few establishing a direct link between international production linkages and changes in the firms' exports and domestic sales during the financial crisis.

Global value chains make firms more vulnerable to external shocks and the following dimensions of firm performance have been considered: Domestic sales growth, foreign sales growth and value-added growth. Several measures of trade exposure at the firm and industry levels were also used. Our main finding points to a positive relationship between trade exposure and firm performance. However, during the crisis, firms with greater exposure to trade experienced a more severe decline in sales and value-added growth. This finding is robust across measures of export exposure. The results also showed that the effect of the crisis on export sales growth was greater for firms that operate in industries that are more reliant on export sales as well as industries that rely on imports. Another important result points to the role of the value chain position. Our results also showed that Swedish industries, that are closer to final demand, reported a larger decline in sales and value-added growth during the crisis.

This paper shows that the coordinated decline in demand across many countries led to adverse outcomes for Swedish manufacturing firms' export and domestic sales growth. One interpretation of these findings is that exporters were less able to re-direct sales from weak markets to stronger ones. The results also point to indirect effects of the crisis as domestic sales decreased during this period. Traditional research on the effect of financial crises on firm performance often focuses on the effect of exchange rate volatility on exports. Our findings show that, in the current era of globalized production, factors such as production linkages may raise the responsiveness of firms to external shocks. One suggestion for future research on the role of trade linkages is to analyze how the decline in exports

influence factors such as the demand for labor. This can improve the understanding of the mechanisms through which the crisis led to a decline in domestic sales.

**Author Contributions:** Conceptualization, T.-A.S.; Data curation, A.M.M.S.Q. and T.-A.S.; Formal analysis, T.-A.S.; Supervision, A.M.M.S.Q.; Validation, A.M.M.S.Q.; Writing—original draft, T.-A.S.; Writing—review & editing, A.M.M.S.Q.

**Funding:** This research received no external funding.

**Conflicts of Interest:** The authors declare no conflict of interest.

## Appendix A

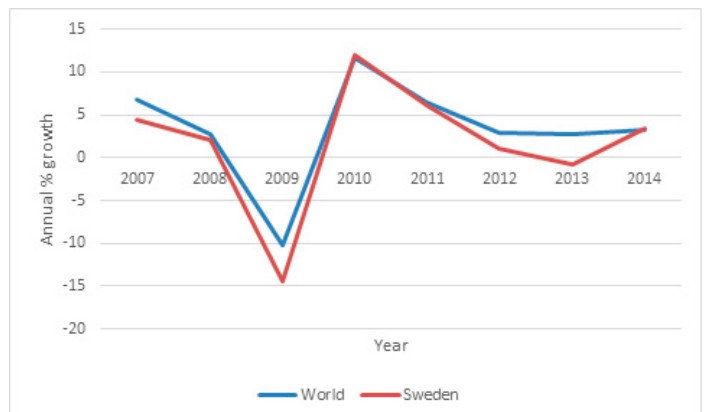

**Figure A1.** Annual percentage growth in trade (2007–2014). Source: World Bank.

**Table A1.** Firm sales and value-added growth.

| Variables | (1) Total Sales | (2) Total Sales | (3) Total Sales | (4) Value Added | (5) Value Added | (6) Value Added |
|---|---|---|---|---|---|---|
| Crisis | 0.017 (0.016) | 0.007 (0.015) | 0.006 (0.016) | −0.017 *** (0.004) | −0.019 *** (0.004) | −0.018 *** (0.004) |
| Import 2007 | 0.009 (0.010) | | | 0.022 *** (0.002) | | |
| Crisis × Import | −0.064 ** (0.026) | | | 0.011 ** (0.006) | | |
| Exporter | −0.137 *** (0.026) | | | −0.032 *** (0.006) | | |
| Exports 2007 | | 0.051 *** (0.012) | | | 0.052 *** (0.003) | |
| Crisis × Exports 2008 | | −0.076 ** (0.030) | | | 0.004 (0.007) | |
| Exposure | | | −0.024 * (0.014) | | | −0.044 *** (0.003) |
| Crisis × exposure 2007 | | | 0.037 (0.035) | | | 0.027 *** (0.007) |
| Wage per worker | 0.084 *** (0.015) | 0.072 *** (0.014) | 0.070 *** (0.017) | 0.030 *** (0.004) | 0.024 *** (0.004) | 0.022 *** (0.004) |
| HHI | −13.294 *** (1.099) | −12.861 *** (0.837) | −13.019 *** (0.909) | −0.760 *** (0.082) | −0.893 *** (0.085) | −0.932 *** (0.098) |
| Constant | 0.715 *** (0.039) | 0.556 *** (0.027) | 0.535 *** (0.029) | 0.094 *** (0.007) | 0.060 *** (0.004) | 0.058 *** (0.004) |
| Observations | 230,165 | 251,690 | 210,356 | 183,709 | 198,648 | 169,988 |
| R-squared | 0.208 | 0.201 | 0.202 | 0.221 | 0.224 | 0.225 |
| INDUSTRY FE | YES | YES | YES | YES | YES | YES |

Note: Lagged dependent variable included but not reported. Robust standard errors clustered at the firm × year level. Reported in parentheses where *** denotes $p < 0.01$, ** $p < 0.05$, * and $p < 0.1$.

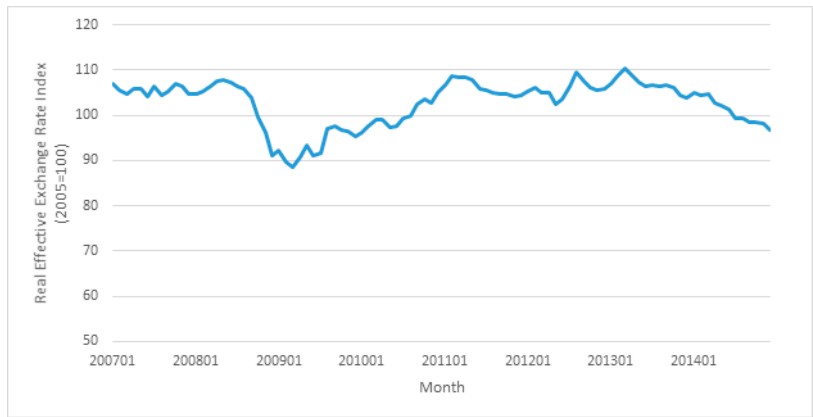

**Figure A2.** Real effective exchange rate index (2007–2014). Source: International Monetary Fund.

**Table A2.** Firm sales and value-added growth.

| Variables | (1) Domestic Sales | (2) Domestic Sales | (3) Domestic Sales | (4) Foreign Sales | (5) Foreign Sales | (6) Foreign Sales |
|---|---|---|---|---|---|---|
| Crisis | 0.023 (0.016) | 0.014 (0.015) | 0.012 (0.016) | 0.008 (0.014) | −0.051 *** (0.015) | −0.039 ** (0.016) |
| Import 2007 | 0.011 (0.010) | | | 0.022 *** (0.008) | | |
| Crisis × Import | −0.065 ** (0.026) | | | −0.018 (0.021) | | |
| Exporter | −0.137 *** (0.026) | | | 0.209 *** (0.008) | | |
| Exports 2007 | | 0.021 * (0.012) | | | 0.202 *** (0.011) | |
| Crisis × Exports 2008 | | −0.033 (0.030) | | | −0.155 *** (0.029) | |
| Exposure | | | −0.052 *** (0.014) | | | 0.067 *** (0.013) |
| Crisis × exposure 2007 | | | 0.070 ** (0.035) | | | −0.045 (0.033) |
| Wage per worker | 0.082 *** (0.015) | 0.072 *** (0.014) | 0.075 *** (0.017) | 0.063 *** (0.011) | 0.079 *** (0.013) | 0.081 *** (0.015) |
| HHI | −13.214 *** (1.091) | −12.779 *** (0.823) | −13.035 *** (0.900) | −8.757 *** (0.747) | −10.756 *** (0.712) | −11.099 *** (0.791) |
| Constant | 0.712 *** (0.039) | 0.540 *** (0.026) | 0.522 *** (0.029) | 0.175 *** (0.022) | 0.508 *** (0.024) | 0.502 *** (0.027) |
| Observations | 230,165 | 251,690 | 210,356 | 230,165 | 251,690 | 210,356 |
| R-squared | 0.209 | 0.203 | 0.204 | 0.215 | 0.201 | 0.200 |
| INDUSTRY FE | YES | YES | YES | YES | YES | YES |

Note: Lagged dependent variable included but not reported. Robust standard errors clustered at the firm × year level. Reported in parentheses where *** denotes $p < 0.01$, ** $p < 0.05$, * and $p < 0.1$.

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
