# Peer review of "Do Global Value Chains Make Firms More Vulnerable to Trade Shocks?—Evidence from Manufacturing Firms in Sweden"

_jrfm, doi:10.3390/jrfm12030151_

Round 1

Reviewer 1 Report

The article is very interesting study on the effect of the global financial crisis from 2008 on manufacturing firms in Sweden. The Authors analysed the direct and indirect effect of trade exposure on firm performance. The methodology is appropriate. The aim of the paper was achieved.

I recommend adding some research question at the beginning of the article and the literature review. Some subject literature was presented in the section "Introduction", but I think that the Authors should add a separate section with the literature review.

To conclude, it is really easy for me to accept this paper as a publishable article (after some revisions) given the following reasons:

(1) The research methods and material used in the paper is adequate to the problem.

(2) The article is with clear goal and methodology.

(3) The article provides new information or new ideas to the subject literature.

Author Response

Reviewer:

I recommend adding some research question at the beginning of the article and the literature review. Some subject literature was presented in the section "Introduction", but I think that the Authors should add a separate section with the literature review.

Answer: We find it difficult to make a separate section for literature review. We have made introduction clearer and better structured on research question and contribution. Updated the introduction with recent literature.

Reviewer 2 Report

Abstract

Please try yo better highlight the research question and the novelty of your research. It should be clear what is your unique contribution to the literature / added value to the literature. 

Introduction & Literature review

Please try to have 2 separate sections: one section with the introduction where you present the research gap and you show how the research gap is transposed into the research question, afterwards you should present the novelty of the research and explain how the research question is transposed into the methodology and your empirical part. You should argue how you enrich the literature, where is the originality of the paper. Furthermore you should briefly present  the research context and give some managerial and theoretical implications. The introduction should end with a clear presentation of the sections of your article.

Than you need to present the literature review which should emerge logically from the structure of your paper and the research question that you investigate.

Lines 25-26 and others: I think it is more logically to cite references chronologically. 

" understanding the relationship1 " the 1 should be a superscript. However it is not proper for research articles to have such superscripts and notes in the text.

Lines 51-53: please try to have paragraphs which consist of several phrases.

"We use data on Swedish firms’ monthly exports, sales and value added for " this is the research context. Ok, but why Swedish firms? Why not Finish ones? What is so interesting about those firms? From which industry?

"There are two competing hypotheses explaining " are they really hypothesis or research directions in the literature?

" A number 104
of studies investigate this hypothesis (Foley and Manova 2015"

a number of studies...but you cite only one! it is not a hypothesis. maybe a line of research

The methodology of the paper should be very clear and it should be clear from the abstract how you structure that and how the research question translates into the methodology and the investigation you conduct. In other words I would expect that you explain in the introduction what you will do in the methodology and empirical part.

You need another section to explain the research context and give some proper arguments why you chose Sweden! And which industry and why you chose that industry. 

Methodology

relation 1 and others: you need first to specify what each symbol present...and only afterwards to construct - build the relationship.

line 138: I would expect that you have here relation 2. ?

"exposure at the industry level " which is that? What symbols? Which is the dummy?

" Herfindahl 169
index to control" why? please give more arguments and also some references!

" define the crisis period as the period from September 2008 to December 2009" why this period of time? Bring more arguments! Cite some references for that!

The discussion developed in the paper are not strong enough. The authors should try to make some more comparisons with other findings and enhance this part.

Conclusions:

how about managerial implications?

Any limitations?

Any future research perspectives?

The paper is interesting, but some major revisions are necessary. 

Author Response

Reviewer

Abstract

Please try yo better highlight the research question and the novelty of your research. It should be clear what is your unique contribution to the literature / added value to the literature. 

Answer: Taken into consideration.

Reviewer

Introduction & Literature review

Please try to have 2 separate sections: one section with the introduction where you present the research gap and you show how the research gap is transposed into the research question, afterwards you should present the novelty of the research and explain how the research question is transposed into the methodology and your empirical part. You should argue how you enrich the literature, where is the originality of the paper. Furthermore you should briefly present  the research context and give some managerial and theoretical implications. The introduction should end with a clear presentation of the sections of your article.

Than you need to present the literature review which should emerge logically from the structure of your paper and the research question that you investigate.

Answer: We find it difficult to make a separate section for literature review. We have made introduction clearer and better structured on research question and contribution. Updated the introduction with recent literature.

Reviewer:

Lines 25-26 and others: I think it is more logically to cite references chronologically. 

Answer: Done

" understanding the relationship1 " the 1 should be a superscript. However it is not proper for research articles to have such superscripts and notes in the text.

Answer: Done

Lines 51-53: please try to have paragraphs which consist of several phrases.

Answer: Re-arranged!

Reviewer

"We use data on Swedish firms’ monthly exports, sales and value added for " this is the research context. Ok, but why Swedish firms? Why not Finish ones? What is so interesting about those firms? From which industry?

Answer: Why Swedish firm? With my sincere apology, I would like to say that I do not have any answer for that. The only motivation is that we are from Sweden. This answer is implied, I guess. The industry is manufacturing. It is mentioned in the title and 15 other places for example see line 93 in the revised version.

Reviewer:

"There are two competing hypotheses explaining " are they really hypothesis or research directions in the literature?

" A number 104
of studies investigate this hypothesis (Foley and Manova 2015"

a number of studies...but you cite only one! it is not a hypothesis. maybe a line of research

Answer: Taken into consideration and corrected thereafter!

Reviewer:

The methodology of the paper should be very clear and it should be clear from the abstract how you structure that and how the research question translates into the methodology and the investigation you conduct. In other words I would expect that you explain in the introduction what you will do in the methodology and empirical part.

Answer: Abstract and introduction is re-written and restructured:

Reviewer:

You need another section to explain the research context and give some proper arguments why you chose Sweden! And which industry and why you chose that industry. 

Answer: Sorry, I do not agree the comments. It is the interest of Swedish people and the people of those countries that Sweden has trade with. This motivation is implied. We dealt with full population, so we do not need to motivate in terms of result generalization or statistical properties.

Reviewer:

Methodology

relation 1 and others: you need first to specify what each symbol present...and only afterwards to construct - build the relationship.

line 138: I would expect that you have here relation 2. ?

"exposure at the industry level " which is that? What symbols? Which is the dummy?

Answer: Re-written and made clear!

" Herfindahl 169 index to control" why? please give more arguments and also some references!

Answer: It is already explained. The term is well established in the literature.

Reviewer:

" define the crisis period as the period from September 2008 to December 2009" why this period of time? Bring more arguments! Cite some references for that!

Answer: Reformulated and reference is given.

Reviewer:

The discussion developed in the paper are not strong enough. The authors should try to make some more comparisons with other findings and enhance this part.

Conclusions:

how about managerial implications?

Any limitations?

Any future research perspectives?

The paper is interesting, but some major revisions are necessary. 

Answer: No comments!

Round 2

Reviewer 2 Report

Congratulations. The authors have implemented most of my suggestions. Good luck with the paper.